# Learning Wake-Sleep Recurrent Attention Models

**Jimmy Ba**
University of Toronto
jimmy@psi.toronto.edu

**Roger Grosse**
University of Toronto
rgrosse@cs.toronto.edu

**Ruslan Salakhutdinov**
University of Toronto
rsalskhu@cs.toronto.edu

**Brendan Frey**
University of Toronto
frey@psi.toronto.edu

## Abstract

Despite their success, convolutional neural networks are computationally expensive because they must examine all image locations. Stochastic attention-based models have been shown to improve computational efficiency at test time, but they remain difficult to train because of intractable posterior inference and high variance in the stochastic gradient estimates. Borrowing techniques from the literature on training deep generative models, we present the Wake-Sleep Recurrent Attention Model, a method for training stochastic attention networks which improves posterior inference and which reduces the variability in the stochastic gradients. We show that our method can greatly speed up the training time for stochastic attention networks in the domains of image classification and caption generation.

## 1 Introduction

Convolutional neural networks, trained end-to-end, have been shown to substantially outperform previous approaches to various supervised learning tasks in computer vision (e.g. [1])). Despite their wide success, convolutional nets are computationally expensive when processing high-resolution input images, because they must examine all image locations at a fine scale. This has motivated recent work on visual attention-based models [2, 3, 4], which reduce the number of parameters and computational operations by selecting informative regions of an image to focus on. In addition to computational speedups, attention-based models can also add a degree of interpretability, as one can understand what signals the algorithm is using by seeing where it is looking. One such approach was recently used by [5] to automatically generate image captions and highlight which image region was relevant to each word in the caption.

There are two general approaches to attention-based image understanding: hard and soft attention. Soft attention based models (e.g. [5]) obtain features from a weighted average of all image locations, where locations are weighted based on a model's saliency map. By contrast, a hard attention model (e.g. [2, 3]) chooses, typically stochastically, a series of discrete glimpse locations. Soft attention models are computationally expensive, as they have to examine every image location; we believe that the computational gains of attention require a hard attention model. Unfortunately, this comes at a cost: while soft attention models can be trained with standard backpropagation [6, 5], this does not work for hard attention models, whose glimpse selections are typically discrete.

Training stochastic hard attention models is difficult because the loss gradient involves intractable posterior expectations, and because the stochastic gradient estimates can have high variance. (The latter problem was also observed by [7] in the context of memory networks.) In this work, we propose the Wake-Sleep Recurrent Attention Model (WS-RAM), a method for training stochastic recurrent attention models which deals with the problems of intractable inference and high-variance gradients by taking advantage of several advances from the literature on training deep generative models:

inference networks [8], the reweighted wake-sleep algorithm [9], and control variates [10, 11]. During training, the WS-RAM approximates posterior expectations using importance sampling, with a proposal distribution computed by an inference network. Unlike the prediction network, the inference network has access to the object category label, which helps it choose better glimpse locations. As the name suggests, we train both networks using the reweighted wake-sleep algorithm. In addition, we reduce the variance of the stochastic gradient estimates using carefully chosen control variates. In combination, these techniques constitute an improved training procedure for stochastic attention models.

The main contributions of our work are the following. First, we present a new learning algorithm for stochastic attention models and compare it with a training method based on variational inference [2]. Second, we develop a novel control variate technique for gradient estimation which further speeds up training. Finally, we demonstrate that our stochastic attention model can learn to (1) classify translated and scaled MNIST digits, and (2) generate image captions by attending to the relevant objects in images and their corresponding scale. Our model achieves similar performance to the variational method [2], but with much faster training times.

## 2 Related work

In recent years, there has been a flurry of work on attention-based neural networks. Such models have been applied successfully in image classification [12, 4, 3, 2], object tracking [13, 3], machine translation [6], caption generation [5], and image generation [14, 15]. Attention has been shown both to improve computational efficiency [2] and to yield insight into the network's behavior [5].

Our work is most closely related to stochastic hard attention models (e.g. [2]). A major difficulty of training such models is that computing the gradient requires taking expectations with respect to the posterior distribution over saccades, which is typically intractable. This difficulty is closely related to the problem of posterior inference in training deep generative models such as sigmoid belief networks [16]. Since our proposed method draws heavily from the literature on training deep generative models, we overview various approaches here.

One of the challenges of training a deep (or recurrent) generative model is that posterior inference is typically intractable due to the explaining away effect. One way to deal with intractable inference is to train a separate inference network whose job it is to predict the posterior distribution. A classic example was the Helmholtz machine [8], where the inference network predicts a mean field approximation to the posterior.[1] The generative and inference networks are trained with the wake-sleep algorithm: in the wake phase, the generative model is updated to increase a variational lower bound on the data likelihood. In the sleep phase, data are generated from the model, and the inference network is trained to predict the latent variables used to generate the observations.

The wake-sleep approach was limited by the fact that the wake and sleep phases were minimizing two unrelated objective functions. More recently, various methods have been proposed which unify the training of the generative and inference networks into a single objective function. Neural variational inference and learning (NVIL) [11] trains both networks to maximize a variational lower bound on the log-likelihood. Since the stochastic gradient estimates in NVIL are very noisy, the method of control variates is used to reduce the variance. In particular, one uses an algortihm from reinforcement learning called REINFORCE [17], which attempts to infer a reward baseline for each instance. The choice of baseline is crucial to good performance; NVIL uses a separate neural network to compute the baseline, an approach also used by [3] in the context of attention networks. Control variates are discussed in more detail in Section 4.4.

The reweighted wake-sleep approach [9] is similar to traditional wake-sleep, but uses importance sampling in place of mean field inference to approximate the posterior. Reweighted wake-sleep is described more formally in Section 4.3. Another method based on inference networks is variational autoencoders [18, 19], which exploit a clever reparameterization of the probabilistic model in order to improve the signal in the stochastic gradients. NVIL, reweighted wake-sleep, and variational autoencoders have all been shown to achieve considerably higher test log-likelihoods compared to

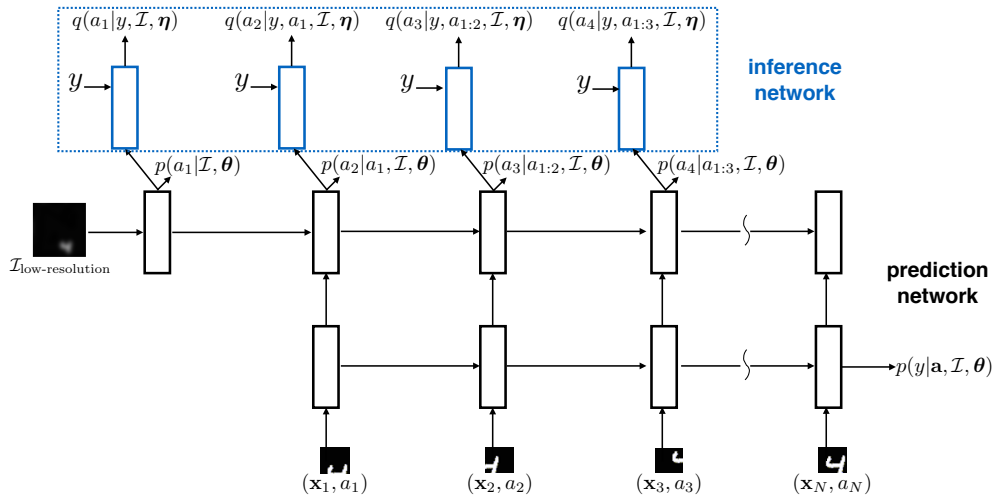

Figure 1: The Wake-Sleep Recurrent Attention Model.

traditional wake-sleep. The term "Helmholtz machine" is often used loosely to refer to the entire collection of techniques which simultaneously learn a generative network and an inference network.

## 3 Wake-Sleep Recurrent Attention Model

We now describe our wake-sleep recurrent attention model (WS-RAM). Given an image $\mathcal{I}$, the network first chooses a sequence of glimpses $\mathbf{a} = (a_1, \ldots, a_N)$, and after each glimpse, receives an observation $\mathbf{x}_n$ computed by a mapping $g(a_n, \mathcal{I})$. This mapping might, for instance, extract an image patch at a given scale. The first glimpse is based on a low-resolution version of the input, while subsequent glimpses are chosen based on information acquired from previous glimpses. The glimpses are chosen stochastically according to a distribution $p(a_n \mid a_{1:n-1}, \mathcal{I}, \boldsymbol{\theta})$, where $\boldsymbol{\theta}$ denotes the parameters of the network. This is in contrast with soft attention models, which deterministically allocate attention across all image locations. After the last glimpse, the network predicts a distribution $p(y \mid \mathbf{a}, \mathcal{I}, \boldsymbol{\theta})$ over the target $y$ (for instance, the caption or image category).

As shown in Figure 1, the core of the attention network is a two-layer recurrent network, which we term the "prediction network", where the output at each time step is an action (saccade) which is used to compute the input at the next time step. A low-resolution version of the input image is fed to the network at the first time step, and the network predicts the class label at the final time step. Importantly, the low-resolution input is fed to the second layer, while the class label prediction is made by the first layer, preventing information from propagating directly from the low-resolution image to the output. This prevents local optima where the network learns to predict $y$ directly from the low-resolution input, disregarding attention completely.

On top of the prediction network is an inference network, which receives both the class label and the attention network's top layer representation as inputs. It tries to predict the posterior distribution $q(a_{n+1} \mid y, a_{1:n}, \mathcal{I}, \boldsymbol{\eta})$, parameterized by $\boldsymbol{\eta}$, over the next saccade, conditioned on the image category being correctly predicted. Its job is to guide the posterior sampler during training time, thereby acting as a "teacher" for the attention network. The inference network is described further in Section 4.3.

One of the benefits of stochastic attention models is that the mapping $g$ can be localized to a small image region or coarse granularity, which means it can potentially be made very efficient. Furthermore, $g$ need not be differentiable, which allows for operations (such as choosing a scale) which would be difficult to implement in a soft attention network. The cost of this flexibility is that standard backpropagation cannot be applied, so instead we use novel algorithms described in the next section.

# 4 Learning

In this work, we assume that we have a dataset with labels $y$ for the supervised prediction task (e.g. object category). In contrast to the supervised saliency prediction task (e.g. [20, 21]), there are no labels for where to attend. Instead, we learn an attention policy based on the idea that the best locations to attend to are the ones which most robustly lead the model to predict the correct category. In particular, we aim to maximize the probability of the class label (or equivalently, minimize the cross-entropy) by marginalizing over the actions at each glimpse:

$$\ell = \log p(y \,|\, \mathcal{I}, \boldsymbol{\theta}) = \log \sum_{\mathbf{a}} p(\mathbf{a} \,|\, \mathcal{I}, \boldsymbol{\theta}) p(y \,|\, \mathbf{a}, \mathcal{I}, \boldsymbol{\theta}). \tag{1}$$

We train the attention model by maximizing a lower bound on $\ell$. In Section 4.1, we first describe a previous approach which minimized a variational lower bound. We then introduce our proposed method which directly estimates the gradients of $\ell$. As shown in Section 4.2, our method can be seen as maximizing a tighter lower bound on $\ell$.

## 4.1 Variational lower bound

We first outline the approach of [2], who trained the model to maximize a variational lower bound on $\ell$. Let $q(\mathbf{a} \,|\, y, \mathcal{I})$ be an approximating distribution. The lower bound on $\ell$ is then given by:

$$\ell = \log \sum_{\mathbf{a}} p(\mathbf{a} \,|\, \mathcal{I}, \boldsymbol{\theta}) p(y \,|\, \mathbf{a}, \mathcal{I}, \boldsymbol{\theta}) \geq \sum_{\mathbf{a}} q(\mathbf{a} \,|\, y, \mathcal{I}) \log p(y, \mathbf{a} \,|\, \mathcal{I}, \boldsymbol{\theta}) + \mathcal{H}[q] = \mathcal{F}. \tag{2}$$

In the case where $q(\mathbf{a} \,|\, y, \mathcal{I}) = p(\mathbf{a} \,|\, \mathcal{I}, \boldsymbol{\theta})$ is the prior, as considered by [2], this reduces to

$$\mathcal{F} = \sum_{\mathbf{a}} p(\mathbf{a} \,|\, \mathcal{I}, \boldsymbol{\theta}) \log p(y \,|\, \mathbf{a}, \mathcal{I}, \boldsymbol{\theta}). \tag{3}$$

The learning rules can be derived by taking derivatives of Eqn. 3 with respect to the model parameters:

$$\frac{\partial \mathcal{F}}{\partial \boldsymbol{\theta}} = \sum_{\mathbf{a}} p(\mathbf{a} \,|\, \mathcal{I}, \boldsymbol{\theta}) \left[ \frac{\partial \log p(y \,|\, \mathbf{a}, \mathcal{I}, \boldsymbol{\theta})}{\partial \boldsymbol{\theta}} + \log p(y \,|\, \mathbf{a}, \mathcal{I}, \boldsymbol{\theta}) \frac{\partial \log p(\mathbf{a} \,|\, \mathcal{I}, \boldsymbol{\theta})}{\partial \boldsymbol{\theta}} \right]. \tag{4}$$

The summation can be approximated using $M$ Monte Carlo samples $\tilde{\mathbf{a}}^m$ from $p(\mathbf{a} \,|\, \mathcal{I}, \boldsymbol{\theta})$:

$$\frac{\partial \mathcal{F}}{\partial \boldsymbol{\theta}} \approx \frac{1}{M} \sum_{m=1}^{M} \left[ \frac{\partial \log p(y \,|\, \tilde{\mathbf{a}}^m, \mathcal{I}, \boldsymbol{\theta})}{\partial \boldsymbol{\theta}} + \log p(y \,|\, \tilde{\mathbf{a}}^m, \mathcal{I}, \boldsymbol{\theta}) \frac{\partial \log p(\tilde{\mathbf{a}}^m \,|\, \mathcal{I}, \boldsymbol{\theta})}{\partial \boldsymbol{\theta}} \right]. \tag{5}$$

The partial derivative terms can each be computed using standard backpropagation. This suggests a simple gradient-based training algorithm: For each image, one first computes the samples $\tilde{\mathbf{a}}^m$ from the prior $p(\mathbf{a} \,|\, \mathcal{I}, \boldsymbol{\theta})$, and then updates the parameters according to Eqn. 5. As observed by [2], one must carefully use control variates in order to make this technique practical; we defer discussion of control variates to Section 4.4.

## 4.2 An improved lower bound on the log-likelihood

The variational method described above has some counterintuitive properties early in training. First, because it averages the log-likelihood over actions, it greatly amplifies the differences in probabilities assigned to the true category by different bad glances. For instance, a glimpse sequence which leads to 0.01 probability assigned to the correct class is considered much worse than one which leads to 0.02 probability under the variational objective, even though in practice they may be equally bad since they have both missed the relevant information. A second odd behavior is that all glimpse sequences are weighted equally in the log-likelihood gradient. It would be better if the training procedure focused its effort on using those glances which contain the relevant information. Both of these effects contribute noise in the training procedure, especially in the early stages of training.

Instead, we adopt an approach based on the wake-$p$ step of reweighted wake-sleep [9], where we attempt to maximize the marginal log-probability $\ell$ directly. We differentiate the marginal log-likelihood objective in Eqn. 1 with respect to the model parameters:

$$\frac{\partial \ell}{\partial \boldsymbol{\theta}} = \frac{1}{p(y \mid \mathcal{I}, \boldsymbol{\theta})} \sum_{\mathbf{a}} p(\mathbf{a} \mid \mathcal{I}, \boldsymbol{\theta}) p(y \mid \mathbf{a}, \mathcal{I}, \boldsymbol{\theta}) \left[ \frac{\partial \log p(y \mid \mathbf{a}, \mathcal{I}, \boldsymbol{\theta})}{\partial \boldsymbol{\theta}} + \frac{\partial \log p(\mathbf{a} \mid \mathcal{I}, \boldsymbol{\theta})}{\partial \boldsymbol{\theta}} \right]. \quad (6)$$

The summation and normalizing constant are both intractable to evaluate, so we estimate them using importance sampling. We must define a proposal distribution $q(\mathbf{a} \mid y, \mathcal{I})$, which ideally should be close to the posterior $p(\mathbf{a} \mid y, \mathcal{I}, \boldsymbol{\theta})$. One reasonable choice is the prior $p(\mathbf{a} \mid \mathcal{I}, \boldsymbol{\theta})$, but another choice is described in Section 4.3. Normalized importance sampling gives a biased but consistent estimator of the gradient of $\ell$. Given samples $\tilde{\mathbf{a}}^1, \ldots, \tilde{\mathbf{a}}^M$ from $q(\mathbf{a} \mid y, \mathcal{I})$, the (unnormalized) importance weights are computed as:

$$\tilde{w}^m = \frac{p(\tilde{\mathbf{a}}^m \mid \mathcal{I}, \boldsymbol{\theta}) p(y \mid \tilde{\mathbf{a}}^m, \mathcal{I}, \boldsymbol{\theta})}{q(\tilde{\mathbf{a}}^m \mid y, \mathcal{I})}. \quad (7)$$

The Monte Carlo estimate of the gradient is given by:

$$\frac{\partial \ell}{\partial \boldsymbol{\theta}} \approx \sum_{m=1}^{M} w^m \left[ \frac{\partial \log p(y \mid \tilde{\mathbf{a}}^m, \mathcal{I}, \boldsymbol{\theta})}{\partial \boldsymbol{\theta}} + \frac{\partial \log p(\tilde{\mathbf{a}}^m \mid \mathcal{I}, \boldsymbol{\theta})}{\partial \boldsymbol{\theta}} \right], \quad (8)$$

where $w^m = \tilde{w}^m / \sum_{i=1}^{M} \tilde{w}^i$ are the normalized importance weights. When $q$ is chosen to be the prior, this approach is equivalent to the method of [22] for learning generative feed-forward networks.

Our importance sampling based estimator can also be viewed as the gradient ascent update on the objective function $\mathbb{E}\left[\log \frac{1}{M} \sum_{m=1}^{M} \tilde{w}^m\right]$. Combining Jensen's inequality with the unbiasedness of the $\tilde{w}^m$ shows that this is a lower bound on the log-likelihood:

$$\mathbb{E}\left[\log \frac{1}{M} \sum_{m=1}^{M} \tilde{w}^m\right] \leq \log \mathbb{E}\left[\frac{1}{M} \sum_{m=1}^{M} \tilde{w}^m\right] = \log \mathbb{E}\left[\tilde{w}^m\right] = \ell. \quad (9)$$

We relate this to the previous section by noting that $\mathcal{F} = \mathbb{E}[\log \tilde{w}^m]$. Another application of Jensen's inequality shows that our proposed bound is at least as accurate as $\mathcal{F}$:

$$\mathcal{F} = \mathbb{E}\left[\log \tilde{w}^m\right] = \mathbb{E}\left[\frac{1}{M} \sum_{m=1}^{M} \log \tilde{w}^m\right] \leq \mathbb{E}\left[\log \frac{1}{M} \sum_{m=1}^{M} \tilde{w}^m\right]. \quad (10)$$

Burda et al. [23] further analyzed a closely related importance sampling based estimator in the context of generative models, bounding the mean absolute deviation and showing that the bias decreases monotonically with the number of samples.

### 4.3 Training an inference network

Late in training, once the attention model has learned an effective policy, the prior distribution $p(\mathbf{a} \mid \mathcal{I}, \boldsymbol{\theta})$ is a reasonable choice for the proposal distribution $q(\mathbf{a} \mid y, \mathcal{I})$, as it puts significant probability mass on good actions. But early in training, the model may have only a small probability of choosing a good set of glimpses, and the prior may have little overlap with the posterior. To deal with this, we train an inference network to predict, given the observations *as well as the class label*, where the network should look to correctly predict that class (see Figure 1). With this additional information, the inference network can act as a "teacher" for the attention policy.

The inference network predicts a sequence of glimpses stochastically:

$$q(\mathbf{a} \mid y, \mathcal{I}, \boldsymbol{\eta}) = \prod_{n=1}^{N} q(a_n \mid y, \mathcal{I}, \boldsymbol{\eta}, a_{1:n-1}). \quad (11)$$

This distribution is analogous to the prior, except that each decision also takes into account the class label $y$. We denote the parameters for the inference network as $\boldsymbol{\eta}$. During training, the prediction network is learnt by following the gradient of the estimator in Eqn. 8 with samples $\tilde{\mathbf{a}}^m \sim q(\mathbf{a} \mid y, \mathcal{I}, \boldsymbol{\eta})$ drawn from the inference network output.

Our training procedure for the inference network parallels the wake-$q$ step of reweighted wake-sleep [9]. Intuitively, the inference network is most useful if it puts large probability density over locations in an image that are most informative for predicting class labels. We therefore train the inference weights $\boldsymbol{\eta}$ to minimize the Kullback-Leibler divergence between the recognition model prediction $q(\mathbf{a} \mid y, \mathcal{I}, \boldsymbol{\eta})$ and posterior distribution from the attention model $p(\mathbf{a} \mid y, \mathcal{I}, \boldsymbol{\theta})$:

$$\min_{\boldsymbol{\eta}} D_{\mathrm{KL}}(p \parallel q) = \min_{\boldsymbol{\eta}} - \sum_{\mathbf{a}} p(\mathbf{a} \mid y, \mathcal{I}, \boldsymbol{\theta}) \log q(\mathbf{a} \mid y, \mathcal{I}, \boldsymbol{\eta}). \tag{12}$$

The gradient update for the recognition weights can be obtained by taking the derivatives of Eq. (12) with respect to the recognition weights $\boldsymbol{\eta}$:

$$\frac{\partial D_{\mathrm{KL}}(p \parallel q)}{\partial \boldsymbol{\eta}} = \mathbb{E}_{p(\mathbf{a} \mid y, \mathcal{I}, \boldsymbol{\theta})} \left[ \frac{\partial \log q(\mathbf{a} \mid y, \mathcal{I}, \boldsymbol{\eta})}{\partial \boldsymbol{\eta}} \right]. \tag{13}$$

Since the posterior expectation is intractable, we estimate it with importance sampling. In fact, we reuse the importance weights computed for the prediction network update (see Eqn. 7) to obtain the following gradient estimate for the recognition network:

$$\frac{\partial D_{\mathrm{KL}}(p \parallel q)}{\partial \boldsymbol{\eta}} \approx \sum_{m=1}^{M} w^m \frac{\partial \log q(\tilde{\mathbf{a}}^m \mid y, \mathcal{I}, \boldsymbol{\eta})}{\partial \boldsymbol{\eta}}. \tag{14}$$

### 4.4 Control variates

The speed of convergence of gradient ascent with the gradients defined in Eqns. 8 and 14 suffers from high variance of the stochastic gradient estimates. Past work using similar gradient updates has found significant benefit from the use of control variates, or reward baselines, to reduce the variance [17, 10, 3, 11, 2]. Choosing effective control variates for the stochastic gradient estimators amounts to finding a function that is highly correlated with the gradient vectors, and whose expectation is known or tractable to compute [10, 24]. Unfortunately, a good choice of control variate is highly model-dependent.

We first note that:

$$\mathbb{E}_{q(\mathbf{a} \mid y, \mathcal{I}, \boldsymbol{\eta})} \left[ \frac{p(\mathbf{a} \mid \mathcal{I}, \boldsymbol{\theta})}{q(\mathbf{a} \mid y, \mathcal{I}, \boldsymbol{\eta})} \frac{\partial \log p(\mathbf{a} \mid \mathcal{I}, \boldsymbol{\theta})}{\partial \boldsymbol{\theta}} \right] = 0, \quad \mathbb{E}_{q(\mathbf{a} \mid y, \mathcal{I}, \boldsymbol{\eta})} \left[ \frac{\partial \log q(\mathbf{a} \mid y, \mathcal{I}, \boldsymbol{\eta})}{\partial \boldsymbol{\eta}} \right] = 0. \tag{15}$$

The terms inside the expectation are very similar to the gradients in Eqns. 8 and 14, suggesting that stochastic estimates of these expectations would make good control variates. To increase the correlation between the gradients and the control variates, we reuse the same set of samples and importance weights for the gradients and control variates. Using these control variates results in the gradient estimates for the prediction and recognition networks, we obtain:

$$\frac{\partial \log p(\mathbf{a} \mid \mathcal{I}, \boldsymbol{\theta})}{\partial \boldsymbol{\theta}} \approx \sum_{m=1}^{M} \left( w^m - \frac{\frac{p(\tilde{\mathbf{a}}^m \mid \mathcal{I}, \boldsymbol{\theta})}{q(\tilde{\mathbf{a}}^m \mid y, \mathcal{I}, \boldsymbol{\eta})}}{\sum_{i=1}^{M} \frac{p(\tilde{\mathbf{a}}^i \mid \mathcal{I}, \boldsymbol{\theta})}{q(\tilde{\mathbf{a}}^i \mid y, \mathcal{I}, \boldsymbol{\eta})}} \right) \frac{\partial \log p(\tilde{\mathbf{a}}^m \mid \mathcal{I}, \boldsymbol{\theta})}{\partial \boldsymbol{\theta}}, \tag{16}$$

$$\frac{\partial D_{\mathrm{KL}}(p \parallel q)}{\partial \boldsymbol{\eta}} \approx \sum_{m=1}^{M} \left( w^m - \frac{1}{M} \right) \frac{\partial \log q(\tilde{\mathbf{a}}^m \mid y, \mathcal{I}, \boldsymbol{\eta})}{\partial \boldsymbol{\eta}}. \tag{17}$$

Our use of control variates does not bias the gradient estimates (beyond the bias which is present due to importance sampling). However, as we show in the experiments, the resulting estimates have much lower variance than those of Eqns. 8 and 14.

Following the analogy with reinforcement learning highlighted by [11], these control variates can also be viewed as reward baselines:

$$b_p = \frac{\frac{p(\mathbf{a} \mid \mathcal{I}, \boldsymbol{\theta})}{q(\mathbf{a} \mid y, \mathcal{I}, \boldsymbol{\eta})} \mathbb{E}_{q(\mathbf{a} \mid y, \mathcal{I}, \boldsymbol{\eta})} [p(y \mid \mathbf{a}, \mathcal{I}, \boldsymbol{\theta})]}{M \cdot \mathbb{E}_{q(\mathbf{a} \mid y, \mathcal{I}, \boldsymbol{\eta})} \left[ \frac{p(\mathbf{a} \mid \mathcal{I}, \boldsymbol{\theta})}{q(\mathbf{a} \mid y, \mathcal{I}, \boldsymbol{\eta})} \mathbb{E}_{q(\mathbf{a} \mid y, \mathcal{I}, \boldsymbol{\theta})} [p(y \mid \mathbf{a}, \mathcal{I}, \boldsymbol{\theta})] \right]} \approx \frac{\frac{p(\tilde{\mathbf{a}}^m \mid \mathcal{I}, \boldsymbol{\theta})}{q(\tilde{\mathbf{a}}^m \mid y, \mathcal{I}, \boldsymbol{\eta})}}{\sum_{i=1}^{M} \frac{p(\tilde{\mathbf{a}}^i \mid \mathcal{I}, \boldsymbol{\theta})}{q(\tilde{\mathbf{a}}^i \mid y, \mathcal{I}, \boldsymbol{\eta})}}, \tag{18}$$

$$b_q = \frac{\mathbb{E}_{p(\mathbf{a} \mid \mathcal{I}, \boldsymbol{\theta})} [p(y \mid \mathbf{a}, \mathcal{I}, \boldsymbol{\theta})]}{M \cdot \mathbb{E}_{p(\mathbf{a} \mid \mathcal{I}, \boldsymbol{\theta})} [p(y \mid \mathbf{a}, \mathcal{I}, \boldsymbol{\theta})]} = \frac{1}{M}, \tag{19}$$

where $M$ is the number of samples drawn for proposal $q$.

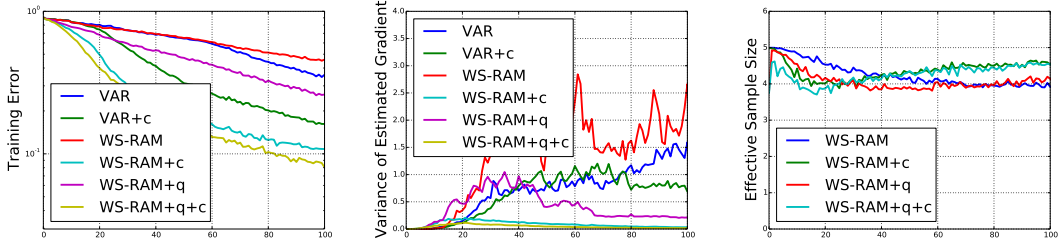

Figure 2: **Left:** Training error as a function of the number of updates. **Middle:** variance of the gradient estimates. **Right:** effective sample size (max = 5). **Horizontal axis:** thousands of updates. **VAR:** variational baseline; **WS-RAM:** our proposed method; **+q:** uses the inference networks for the proposal distribution; **+c:** uses control variates.

### 4.5 Encouraging exploration

Similarly to other methods based on reinforcement learning, stochastic attention networks face the problem of encouraging the method to explore different actions. Since the gradient in Eqn. 8 only rewards or punishes glimpse sequences which are actually performed, any part of the space which is never visited will receive no reward signal. [2] introduced several heuristics to encourage exploration, including: (1) raising the temperature of the proposal distribution, (2) regularizing the attention policy to encourage viewing all image locations, and (3) adding a regularization term to encourage high entropy in the action distribution. We have implemented all three heuristics for the WS-RAM and for the baselines. While these heuristics are important for good performance of the baselines, we found that they made little difference to the WS-RAM because the basic method already explores adequately.

## 5 Experimental results

To measure the effectiveness of the proposed WS-RAM method, we first investigated a toy classification task involving a variant of the MNIST handwritten digits dataset [25] where transformations were applied to the images. We then evaluated the proposed method on a substantially more difficult image caption generation task using the Flickr8k [26] dataset.

### 5.1 Translated scaled MNIST

We generated a dataset of randomly translated and scaled handwritten digits from the MNIST dataset [25]. Each digit was placed in a 100x100 black background image at a random location and scale. The task was to identify the digit class. The attention models were allowed four glimpses before making a classification prediction. The goal of this experiment was to evaluate the effectiveness of our proposed WS-RAM model compared with the variational approach of [2].

For both the WS-RAM and the baseline, the architecture was a stochastic attention model which used ReLU units in all recurrent layers. The actions included both continuous and discrete latent variables, corresponding to glimpse scale and location, respectively. The distribution over actions was represented as a Gaussian random variable for the location and an independent multinomial random variable for the scale. All networks were trained using Adam [27], with the learning rate set to the highest value that allowed the model to successfully converge to a sensible attention policy.

The classification performance results are shown in Table 1. In Figure 2, the WS-RAM is compared with the variational baseline, each using the same number of samples (in order to make computation time roughly equivalent). We also show comparisons against ablated versions of the WS-RAM where the control variates and inference network were removed. When the inference network was removed, the prior $p(\mathbf{a} \,|\, \mathcal{I}, \boldsymbol{\theta})$ was used for the proposal distribution.

In addition to the classification results, we measured the effective sample size (ESS) of our method with and without control variates and the inference network. ESS is a standard metric for evaluating importance samplers, and is defined as $1/\sum_m (w^m)^2$, where $w^m$ denotes the normalized importance weights. Results are shown in Figure 2. Using the inference network reduced the variances in

| Test err. | VAR | WS-RAM | WS-RAM + q |
|-----------|------|--------|------------|
| no c.v.   | 3.11% | 4.23% | 2.59% |
| +c.v.     | 1.81% | 1.85% | 1.62% |

Table 1: Classification error rate comparison for the attention models trained using different algorithms on translated scaled MNIST. The numbers are reported after 10 million updates using 5 samples.

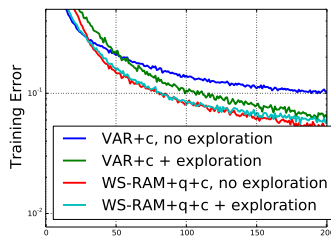

Figure 3: The effect of the exploration heuristics on the variational baseline and the WS-RAM.

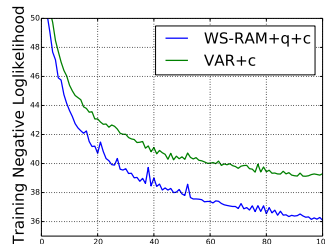

|            | BLEU1 | BLEU2 | BLEU3 | BLEU4 |
|------------|-------|-------|-------|-------|
| VAR        | 62.3  | 41.6  | 26.9  | 17.2  |
| WS-RAM+Qnet | 61.1 | 40.4  | 26.9  | 17.8  |

Table 2: BLEU score performance on the Flickr8K dataset for our WS-RAM and the variational method.

Figure 4: Training negative log-likelihood on Flickr8K for the first 10,000 updates. See Figure 2 for the labels.

gradient estimation, although this improvement did not reflect itself in the ESS. Control variates improved both metrics.

In Section 4.5, we described heuristics which encourage the models to explore the action space. Figure 3 compares the training with and without these heuristics. Without the heuristics, the variational method quickly fell into a local minimum where the model predicted only one glimpse scale over all images; the exploration heuristics fixed this problem. By contrast, the WS-RAM did not appear to have this problem, so the heuristics were not necessary.

## 5.2 Generating captions using multi-scale attention

We also applied the WS-RAM method to learn a stochastic attention model similar to [5] for generating image captions. We report results on the widely-used Flickr8k dataset. The training/valid/test split followed the same protocol as used in previous work [28].

The goal of this experiment was to examine the improvement of the WS-RAM over the variational method for learning with realistic imgaes. Similarly to [5], we first ran a convolutional network, and the attention network then determined which part of the convolutional net representation to attend to. The attention network predicted both which layer to attend to and a location within the layer, in contrast with [5], where the scale was held fixed. Because a convolutional net shrinks the representation with max-pooling, choosing a layer is analogous to choosing a scale. At each glimpse, the inference network was given the immediate preceding word in the target sentences. We compare the BLEU scores of our WS-RAM and the variational method in in Table 2. Figure 4 shows training curves for both models. We observe that WS-RAM obtained similar performance to the variatinoal method, but trained more efficiently.

## 6 Conclusions

In this paper, we introduced the Wake-Sleep Recurrent Attention Model (WS-RAM), an efficient method for training stochastic attention models. This method improves upon prior work by using the reweighted wake-sleep algorithm [9] to approximate expectations from the posterior over glimpses. We also introduced control variates to reduce the variability of the stochastic gradients. Our method reduces the variance in the gradient estimates and accelerates training of attention networks for both invariant handwritten digit recognition and image caption generation.

**Acknowledgments**
This work was supported by the Fields Institute, Samsung, ONR Grant N00014-14-1-0232 and the hardware donation of NVIDIA Corporation.

## Footnotes

[1]In the literature, the inference network is often called a recognition network; we avoid this terminology to prevent confusion with the task of image classification.

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
