[Reviews · NeurIPS 2015]

Submitted by Assigned_Reviewer_1

This paper presents a new learning algorithm for stochastic attention models, employing some recent advances from the literature on training generative models: specifically (1) a simultaneously trained inference network; (2) importance sampling to estimate gradients (a la "reweighted wake sleep"); and (3) tailored control-variates to reduce estimator variance.

The use of control variates and (to a lesser degree) a trained inference network are shown to give significant speed-ups to training time.

Using this learning algorithm, the model is shown to be capable of achieving comparable performance to a recent variational approach [Mnih et al. ICLR2015] (in it's naive form), but with shorter training times.

The authors commendably pack a lot of detail into a small number of pages, and the overall body of work is impressive.

However, I was left with some minor questions (see below) about details of the results -- which hopefully the authors can address in their feedback.

== Overall the paper has good quality, and it uses a nice combination of sophisticated techniques to achieve good results on interesting problem domains.

Equation (10) was also informative/interesting; in the limit, the proposed method gives at least as accurate a bound on the objective as previous variational approaches.

One area of confusion that I have is in regards to the performance in comparison to the variational method of Mnih et al, with respect to modifications on top of that model that encourage exploration. It seems that when compared that the variational approach *without* exploration the proposed method shows a significant speed-up. However, with Figure 3 it appears the the method of Mnih et al *with* exploration significantly out-performs (in terms of speed) the proposed approach (and the proposed approach does not seem to benefit from exploration). Is this the case? And if so, why might this be the case, and are there analogous exploration-boosting methods that one could/should consider for the case of the Flickr 8K example in Figure 4? Granted space is limited -- but it would be useful if the authors could address these issues more thoroughly.

Also, for the learning curves presented, it might also be informative to show testing/validation error progress in addition (or instead of) training progress.

-- In terms of clarity, the paper is reasonably clear and well written. However, there are several parts that could be improved in this regard (in addition to the questions above).

One general comment here is attention to detail, and precision with notation. There are several parts of section for in particular where notation usage is a little sloppy. With a little thought from the reader, it's possible to figure out the precise meanings but it would be useful if the authors could make everything explicit. (As an example of this, the expectation distribution in (9) and (10) is not specified, however it is in (13), (15), etc. Another example is the use of set notation {a_n} and an unadorned "a" symbol both represent a sequence of actions.

Similarly, it's possible to infer the legends of the figures but this could be made clearer and more explicit in several cases.

-- This work has moderate originality. It is a sensible and non-trivial combination of models and techniques that have been proposed recently in the literature.

-- Likewise, I think the work has moderate significance. It represents (potentially) algorithmic improvements/speed-ups to an interesting model, using techniques that are currently gaining favour in the community. It remains to be seen whether the methods proposed here have real-world impact (improvements) any significant applications, and whether they help consistently move forward the state of the art. It does, however, represent a useful exploration of the overall model/algorithm design space and in this sense I found it quite valuable.
Summary: This was an interesting paper using recent advances from training generative models (most directly reweighted-wake-sleep) with the goal of improving training of stochastic-attention models. The main result is that by employing importance sampling and control variates, one can gain significant speed-ups over a more naive variational approaches to the same problem.

Submitted by Assigned_Reviewer_2

This paper presents stochastic attention-based recurrent networks for structured prediction tasks. The proposed recurrent attention model consists of two layers: one inference network that draws glimpses (hard attention) from the input image and one generative network that makes prediction. Compared to variational methods, this paper considers to directly maximize log likelihood with importance sampling that leads to an improved lower bound on log likelihood. The training procedure is thus based on the reweighted wake-sleep algorithm [9]. To train the inference network, this paper defines a proposal distribution for drawing recurrent glimpses conditioned on both labels and observations. To manage the high variance of stochastic gradients, this paper uses control variates in previous literature. Experiments are carried out on a variant of MNIST for classification and the Flickr8k dataset for caption generation. This paper mainly compares with a variational method for training the network, and analyzes the effects of using the new proposal distribution and control variates.

Pros: + This paper presents a nice combination of ideas for training deep generative models, i.e. reweighted wake-sleep and control variates for the proposed recurrent attention model.

+ I believe the main novelty lies the proposal probability that involves class labels, which reduces the randomness of glimpse prediction in the early stage of training procedure. Its effectiveness is demonstrated in Figure 2, 3 and 4.

+ The paper is in general well written and insightful.

Cons:

- Control variates seem helpful for both variational methods and the proposed importance sampling based. It will be interesting to see whether the proposal probability in this paper could be used for variational methods.
Summary: This is a nice work. It combines nicely previous ideas with a novel one for training stochastic recurrent attention models. The proposed inference network is both theoretically and empirically convincing.

Submitted by Assigned_Reviewer_3

Summary: The paper presents a recurrent attention model similar to that in [2], and presents a training algorithm by taking advantage of several advances from the literature on training deep generative models. Empirical results show that the model achieves similar performance to [2] but with less training time.

Quality: Good.

Clarity: The description of the proposed model is unclear. It is unclear how the model depicted in Figure 1 is actually implemented in experiments. For example, what do the white rectangles represent? What subnetwork is used to extract image features? What subnetwork is used as the "two-layer recurrent network"? By saying "recurrent network", do you mean the weights represented by horizontal arrows shared? The description of the model is too abstract to understand. I suggest the authors at least put some equations explaining the input and output of each block in the model. That would help readers understand the model more clearly.

Originaliy: somehow original.

Significance: Not so much.

Since this model is similar to the model in [2], the conclusion would be more convincing by comparing the performances on the challenging dataset studie in [2], that is, SVHN.

A small problem: Line 366: "corresponding to glimpse scale and location, respectively" is incorrect.

Summary: The paper presents an improved training algorithm for a stochastic attention-based model. The results are good.

Author Feedback
Author rebuttal: We thank all of the reviewers for their valuable comments and their detailed suggestions on how to improve the paper. We will make all of the recommended changes. As Reviewers 1 and 3 recommend, we will improve the consistency of the notation. We will also modify the figures and text to better explain the model architecture.

Reviewers 1 and 6 ask about curves for validation error. The test and validation curves follow a very similar trend to the training curves. We will include these curves in the final version.

Reviewer 2 suggests combining the inference network with the variational method. This will make an interesting comparison, and we will include it in the final version.

Reviewer 3 asked how our MNIST architecture relates to that of [2]. In addition to our algorithmic contributions, the network architecture is also more powerful in that it includes a discrete scale variable in addition to the continuous-valued location. The combination of continuous and discrete variables makes the model hard to learn. To the best of our knowledge, our work is the first successful attempt to learn a hard attention model with both continuous and discrete random variables. (Note that obtaining the computational benefits of a multi-resolution attention model requires hard attention, since a soft attention model must examine all locations and scales.)

About the level of challenge of our datasets: our second set of experiments concerned caption generation, which is still an active and ambitious area of research. Our method outperformed a variational attention model which was presented at ICML one month ago. This demonstrates that our method can improve performance on a modern architecture applied to a challenging dataset.

Reviewer 1 pointed out the exploration effect in figure 3 for the variational method. 'var + exploration' was mistakenly plotted with half the length scale of the others. In the correct plot, the 'var+c+exploration' model reaches approximately the same training error as the WS-RAM at epoch 200, but does so slightly more slowly.